# Uncontrolled Donation after Circulatory Death Only Lung Program: An Urgent Opportunity

**DOI:** 10.3390/jcm12206492

**Published:** 2023-10-12

**Authors:** Chiara Lazzeri, Manuela Bonizzoli, Simona Di Valvasone, Adriano Peris

**Affiliations:** Intensive Care Unit and Regional ECMO Referral Center Emergency Department, Azienda Ospedaliero-Universitaria Careggi, Largo Brambilla 3, 50134 Florence, Italyadriano.peris@gmail.com (A.P.)

**Keywords:** uncontrolled donation after circulatory death, lung program, lung maintenance, outcome

## Abstract

Uncontrolled donation after circulatory death (uDCD) represents a potential source of lungs, and since Steen’s 2001 landmark case in Sweden, lungs have been recovered from uDCD donors and transplanted to patients in other European countries (France, the Netherlands, Spain and Italy) with promising results. Disparities still exist among European countries and among regions in Italy due to logistical and organizational factors. The present manuscript focuses on the clinical experiences pertaining to uDCD lungs in North America and European countries and on different lung maintenance methods. Existing experiences (and protocols) are not uniform, especially with respect to the type of lung maintenance, the definition of warm ischemic time (WIT) and, finally, the use of ex vivo perfusion (available in the last several years in most centers). In situ lung cooling may be superior to protective ventilation, but this process may be difficult to perform in the uDCD setting and is also time-consuming. On the other hand, the “protective ventilation technique” is simpler and feasible in every hospital. It may lead to a broader use of uDCD lung donors. To date, the results of lung transplants performed after protective ventilation as a preservation technique are scarce but promising. All the protocols comprise, among the inclusion criteria, a witnessed cardiac arrest. The detectable differences included preservation time (240 vs. 180 min) and donor age (<55 years in Spanish protocols and <65 years in Toronto protocols). Overall, independently of the differences in protocols, lungs from uDCD donors show promising results, and the possibility of optimizing ex vivo lung perfusion may broaden the use of these organs.

## 1. Introduction

A shortage of lung donors poses a large obstacle for lung transplantation (LTx) [1]. LTx using lungs obtained from donations after circulatory death (DCD) is now actively implemented in Europe and North America, and the use of DCD donors has been reported to increase the current donor pool by up to 50% [2].

Uncontrolled DCD (uDCD) [3] represents a potential source of lungs and, since Steen’s 2001 landmark case in Sweden [4], lungs have been recovered from uDCD donors and transplanted to patients in other European countries (France, The Netherlands, Spain and Italy) with promising results. Disparities still exist among European countries and among regions in Italy due to logistical and organizational factors.

The present manuscript focuses on the clinical experiences pertaining to uDCD lungs in North American and European countries and on different lung maintenance methods. The aim is to highlight the limits and potentials of an “uDCD only lung program”, since the latter can be theoretically implemented in all hospitals, even in those without extracorporeal cardiopulmonary resuscitation (eCPR).

A PubMed search for “lung transplantation and uncontrolled donation after circulatory death” was performed (only in the English language), including case series and case reports.

The hesitation for the use of lungs through uDCD stems mainly from medical concerns over secondary injury from prolonged warm ischemia. We therefore briefly summarized the main features of lung physiology.

## 2. Relevant Sections

### 2.1. Lung Physiology

The lung shows unique physiology [5] since it is not dependent on blood perfusion for aerobic metabolism, but instead on passive diffusion through the alveoli for oxygen delivery. In contrast with other organs, the lung has relatively low metabolic needs and is privileged by a local storage of oxygen in the alveoli. This concept was supported by Egan et al. [5] who demonstrated experimentally the feasibility of transplanting donor lungs after cardiac arrest and the absence of circulation for up to 2 h. The importance of lung inflation and intra-alveolar oxygen concentration in a DCD setting was further demonstrated by the same group [6], in agreement with animal studies [7]. Adequate gas exchange has been shown to occur after 2 h of warm ischemic time (WIT) in the absence of lung circulation [5,8,9,10], and this warm ischemia time could be increased to 4 h if the donor was heparinized [10]. That is why most programs recommend a WIT < 150 min and a maximum preservation time <240 min [11,12].

Lungs from uDCD may offer some advantages. While brain death is associated with the up-regulation of innate immunity in the donor and a “cytokine storm”, which is known to contribute to lung injury together with mechanical ventilation, lungs recovered from uDCD donors do not undergo such injuries. Inflammatory response has been considered a potential mechanism underlying lung injury in brain death donors [12]. Several investigations in animal models identified that various inflammatory cytokines are elevated in serum following brain death [13]. Bronchoalveolar lavage samples collected from donor lungs following non-traumatic brain injury documented that higher levels of IL-8 and GROa were found compared to living controls [14]. The IL-6/IL-10 ratio in the donor lungs was associated with the development of chronic lung allograft dysfunction [15]. All these findings highlight the importance of inflammatory activation on determining lung injury. In brain death donors, the lung is known to also be susceptible to so-called neurogenic pulmonary edema, caused by both hemodynamic and sympathetic perturbations induced by the “catecholamine storm” [16].

An investigation comparing DCD and DBD lungs revealed specific donor mechanisms. When comparing gene expression levels, inflammation was found to significantly differ between DCD lungs [17]. When examining a larger series of 177 DBD and 65 DCD cases, lung inflammation was found to have increased in DBD lungs, while the activation of cell death pathways increased in DCD lungs [18].

In uDCD (unlike brain death donors and controlled DCD), lung function cannot be assessed before death, but viability can be properly evaluated afterward by means of ex vivo perfusion. Beyond ex vivo perfusion, other strategies are under investigation with regard to the prevention and treatment of lung ischemia-reperfusion injury [19]. In animal models, inhaled beta2adrenoceptor agonists have been reported to have protective effects against ischemia reperfusion injury, probably due to the maintenance of cAMP and adenine nucleotide levels. Some inhaled gases also seem to have beneficial effects. Among these, inhaled nitric oxide may affect pulmonary post-transplantation dysregulation, although with controversial results. Isoflorane pretreatment seems to reduce lung ischemia reperfusion injury, mainly by the inhibition of the induction of cell apoptosis.

### 2.2. Clinical Experiences in North American and European Countries for uDCD Only Lung Programs

Clinical experience in North American and European countries for “uDCD only lung programs” are reported in Table 1. To date, experiences with “uDCD only lung programs” have been small, constituted mainly by case series and mostly from centers in Spain. Nevertheless, in the last several years, barriers regarding uDCD lung programs have been overcome in other centers, thereby increasing the global experience.

In Spain, the first case series was described by Gamez et al. (2005) [11] who reported five lung transplants (four bipulmonary and one unipulmonary) successfully performed from uDCD with good short and mid-term results.

In 2007, Gomez de Antonio et al. [20] reported their experience, which included 17 out-of-hospital non-heart-beating donor (NHBD) lung transplantations performed since 2002 to 2005. In their paper, the authors reported that, since their first NHBD lung transplant, they had been able to use 12% to 13% of all NHBD lungs per year, which, until present, has remained a promising percentage of potential lung transplants. In this case series, the outcomes were deemed encouraging, with 82% survival rates at 3 months, 69% at 1 year and 58% at 3 years. The 3-year survival rate was lower than that reported in the 2005 International Society for Heart and Lung Transplantation (ISHLT), but this datum comprises causes which are not directly related to the procedure since one patient died of air embolism and another from fulminant hepatitis C. Interestingly, acute aortic dissection was the cause of death in one donor and head trauma in another. During the study period (2002–2005), 54 NHBDs were offered (about 18 per year) and 17 NHBDs were accepted (31% acceptance rate), with macroscopic appearance being the most frequent reason for refusal. In those years, ex vivo perfusion was not yet available.

A few years later (2011), Rodriguez et al. [21] reported their historical cohort study of 33 lung transplant recipients with 32 uDCD donors enrolled from 2002 to 2008. Despite the low number of patients, the authors observed that the median of the total ischemia times was longer in the recipients who died (828 vs. 695; *p* = 0.036).

A statistically significant association of mortality with ischemic times and with primary graft dysfunction was observed by Gomez de Antonio et al. [20] who performed a prospective collection of data from all lung transplants from uDCD donors between 2002 and December 2009. Twenty-nine lung transplants were performed with an overall hospital mortality rate of 17% (five patients).

A good patient survival rate (100% after one month and 80% after one year) was observed by Minambers et al. [22] in a small series of five lung transplantations from uDCD donors.

In 2019, Suberviola et al. [23] presented the results of their uDCD lung program which ran since 2012. In this series, only lungs were preserved, with a reduction in WIT (defined as the time from cardiac arrest until the filling of both hemithoraxes with Perfadex). A high percentage of utilized lungs (77%) was observed, in contrast to previous reports [25,26]. Ex vivo lung perfusion has not been available since 2017, so only two lungs were submitted to EVLP. Nevertheless, the survival rates were promising (the 1-month, 1-year and 5-year survival rates were 100%, 87.5% and 87.5%, respectively). A peculiarity of the program described by Suberviola [23] was that the whole process was performed in the ICU.

The same group [27] compared the outcomes of 38 recipients of lungs from uDCD donors with those of 292 recipients of lungs from donors after brain death (DBD) (2002–2012). Early and long outcomes were not different between the two groups (ICU and hospital stay, primary graft dysfunction and chronic graft dysfunction), but significant differences were found in global survival at 1, 5 and 10 years (71.1%, 50.8% and 16.5% versus 75%, 58.4% and 38.1%, resectively; *p* = 0.048). Despite these differences, the results obtained with uDCD lungs were considered acceptable. EVLP was performed in 21% of uDCD lungs (eight cases).

In recent years, additional cases have been published by Suzuki et al. [28] and Valenza et al., all of whom performed with ex situ machine perfusion before transplantation [29].

The peculiarity of the case report published in 2016 by Valenza et al. [29] was that lung preservation was performed by recruitment maneuvers, continuous positive airway pressure, and protective mechanical ventilation, followed by ex vivo lung perfusion (EVLP), while in the Spanish experience, lung maintenance was performed by topical cooling. The rationale for this type of lung maintenance may rely on the results of experimental models, which documented that the prevention of alveolar collapse appears to be the critical factor in protecting the warm ischemic lung from reperfusion injury independent of a continuous oxygen supply [28,29,30,31,32,33].

In 2014, in Milan, a lung DCD project was implemented, including in situ preservation with normothermic open-lung approach and ex situ assessment with ex vivo lung perfusion (EVLP) [29]. A case of uncontrolled DCD lungs successfully treated with an exceptionally prolonged EVLP was reported. Because the donor’s blood count and liver biopsy showed signs of possible leukemia, EVLP was protracted up to 17 h while waiting for immunohistochemical analyses to rule out this diagnosis; eventually, the results were negative, and the lungs were judged suitable for transplantation. The patient was extubated after 36 h and was discharged 21 d after the operation. Despite early recolonization by Pandoraea pnomenusa and airway complications requiring pneumatic dilatation, the patient is alive and has a satisfactory respiratory function 15 months after transplantation [31].

In Canada, Healey et al. [24] reported their experience with lung uDCD, using a simple method for preservation, that is, lung inflation was implemented using a continuous positive airway pressure (CPAP) of 20 cm H_2_O and FiO_2_ of 50%. Five lung transplants were performed with 0% mortality at one month.

Venema et al. described the implementation of a uDCD program for lungs and kidneys in the Netherlands [34]. Although they failed their goal to increase the number of transplantable organs, factors responsible for this phenomenon were documented such as regional feasibility and donor legislation. The authors hypothesized that a prehospital approach consisting of transferring deceased OHCA patients for the sole purpose of donation could overcome these difficulties.

## 3. Discussion

Existing experiences (and protocols) are not uniform, especially with respect to the type of lung maintenance, the definition of warm ischemic time (WIT) and, finally, the use of ex vivo perfusion (available in the last years in most centers).

Table 2 summarizes the two main lung maintenance techniques performed in uDCD lung programs. The type of lung in vivo maintenance does affect the time of organ retrieval, causing the results of the investigations to be hardly comparable.

The safe duration of human lung ischemia is unknown. In animal models, ventilation provides better function after transplantation than unventilated ischemia. In situ lung cooling may be superior to ventilation [34], but this may not be easy to perform in the uDCD setting and is also time-consuming. On the other hand, the “protective ventilation technique” is simpler and feasible in every hospital. It may lead to a broader use of uDCD lung donors. To date, the results of lung transplants performed after protective ventilation as a preservation technique are scarce but promising [24,29,34,35].

### Definition of Warm Ischemic Time (WIT)

The majority of experimental data suggest that lungs remain viable for at least 60 to 90 min after circulatory arrest [36,37,38,39,40,41,42].

Warm ischemic time was defined by the Spanish group as the time from cardiac arrest to topical cooling and a maximum of 120 min was adopted, as an arbitrary cut-off (90 min from cardiac arrest to hospital arrival, with an additional 30 min to start the preservation maneuvers) [11,12,23,26]. The maximum permitted preservation time was 240 min (from topical cooling to implantation) in protocols by Gamez et al. and Gomez de Antonio (2007) and 180 min by Suberviola et al. [11,12,23,26]. The total ischemic time was defined as the time from cardiac arrest to recipient reperfusion (first and second lungs).

Palleschi et al. defined warm ischemic time from cardiac arrest to pulmonary flushing and total preservation time (from the end of resuscitation until reperfusion of the first lung) in their work [33].

All protocols, independent of countries, comprised, among inclusion criteria, a witnessed cardiac arrest. The detectable differences included preservation time (240 vs. 180 min) and donor age (<55 years in Spanish protocols and <65 years in Toronto protocols). Increasing evidence suggests that ex vivo perfusion (ex vivo lung perfusion EVLP) allows for a better assessment of lung function and quality. Consensus on the best protocol for EVLP is yet to be reached, but research in this filed is ongoing. One of the topics of ongoing research are re-assessing the optimal temperature for static cold storage. The use of ex vivo lung perfusion machine as an immunoregulating tool for inducing better tolerance in the recipient after transplant is another promising topic of research.

## 4. Conclusions

Uncontrolled DCD (uDCD) represents a potential source of lungs. Existing experiences (and protocols) are not uniform, especially with respect to the type of lung maintenance, the definition of warm ischemic time (WIT) and, finally, the use of ex vivo perfusion (available in the last few years in most centers). The two main lung maintenance techniques performed in uDCD lung programs are topical cooling (Spanish group) and protective ventilation. Independently of the differences in protocols, lungs from uDCD donors show promising results, and the possibility of optimizing ex vivo lung perfusion may broaden the use of these organs.

## Figures and Tables

**Table 1 jcm-12-06492-t001:** Investigations of uDCD lung programs.

	Regions	Study Population	Family Consent	Results	Mean Warm Ischemic Time	Utilization Rate	Outcome
Spain							
Gomez de Antonio et al. (2007) [20]	Madrid	17 uDCD donors	Not available	54 effective uDCD, 17 lung transplants	Mean warm ischemic time was 118 min (95% confidence interval ((CI), 44–192 min), total ischemic time was 586 min (95% CI, 402–770 min)—first lung	17/54, 31%	
Rodriguez et al. (2011) [21]	Madrid	78 potential uDCD donors; 32 effective uDCD donors; 26 actual uDCD donors		Recipients had 30 day mortality: 4 (12.1%)	Median of total ischemia times longer in the recipients who died (828 vs. 695; *p* = 0.036).	26/30, 86%	
Minambres et al. (2015) [22]	Santander	11 potential LT uDCD donors; 7 effective uDCD donors; 5 actual uDCD donors				5/7, 71%	The lung transplant patient survival rate was 100% after one month and 80% after one year.
Suberviola et al. (2019) [23]	Santander	22 potential uDCD donors; 9 effective uDCD donors; 7 actual uDCD donors			Mean total ischemic time was 678 min	7/9, 77.7%	The 1-month, 1-year and 5-year survival rates were 100%, 87.5% and 87.5%, respectively. Mean follow-up was 52 months.
Canada							
Healey et al. (2020) [24]	Toronto	44 potential uDCD donors	30 uDCD (68%)	14 effective uDCD; 5 lungs transplanted (16.7% use rate from consented donors)	The mean warm ischemic time was 2.8 h	5/14, 36%	The 30-day mortality was 0%. Four of 5 patients are alive at a median of 651 days (range: 121–1254 days) with preserved lung function.

**Table 2 jcm-12-06492-t002:** Lung maintenance techniques.

	In Vivo Lung Maintenance	Lung Maintenance in the Operating Room	Ex Vivo Perfusion
Protective Ventilation Technique
Healey et al. (2020)—Toronto protocol [24]	Lung inflation was implemented using a 20 cm continuous positive airway pressure (CPAP) and 50% H_2_O and FiO_2_. The donor was then moved to the operating room and connected to	a ventilator using a tidal volume of 7 mL/kg, 50% FiO_2_ and a 5 cm positive end-expiratory pressure of H_2_O.	EVLP system for 3 to 5 h for the assessment of lung function and quality
Valenza et al. (2016) [28]	A recruitment maneuver after death declaration; ventilated potential donors with a low rate (four breaths per minute) and a very low tidal volume (6 mL/kg).		EVPL
Palleschi et al. (2021) [33]	After death certification, a new recruitment maneuver is performed along with an in situ preservation with protective ventilation (6 mL/kg tidal volume (TV) of ideal body weight, 8 cm positive end-expiratory pressure of H_2_O, 4 bpm respiratory rate (RR), 100% FiO_2_).		EVLP
Topical Cooling
Suberviola et al. (2019) [23]; Gomez de Antonio 2007 [20]—the Spanish protocol	Topical lung cooling through chest tubes; a 24 Fr tube is inserted into each hemithorax (anterior second intercostal space) and Perfadex solution (Medisan, Uppsala, Sweden) is instilled at 4 °C for topical cooling. The orotracheal tube is left open to the exterior. Topical ice is applied to the chest and returns to cool the body. Esophageal temperature is maintained at 20–21 °C.	The topical cooling preservation solution is drained from both pleural cavities, and a 100% fraction of inspired oxygen with 5 cm of H_2_O positive end-expiratory pressure lung ventilation is started. As the lungs are cooled, initial ventilation is applied with a low respiratory rate and tidal volume of 3 mL/kg in order to avoid vessel damage; the tidal volume is later increased slightly.

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
