# Peer review of "Uncontrolled Donation after Circulatory Death Only Lung Program: An Urgent Opportunity"

_jcm, 2023, doi:10.3390/jcm12206492_

Round 1
Reviewer 1 Report
Uncontrolled donation after circulatory death only lung program: an emergency opportunity
Major Comments
This is a simple but powerful review that advances the call to expand uDCD lung donation and transplantation programs.
I only have minor comments below.
You say that disparities still exist due to logistical and organization factors – but provide no evidence for this claim.
Either argue why this may be the case or delete the claim and just present the review. Or ask this as a question in the discussion – why is uDCD not expanding and being accepted more rapidly?
What do you as authors hope from this paper? Do you have a call to action?
Minor Comments
Typos/grammar
Abstract
Better: ‘Uncontrolled DCD (uDCD) represents a potential source of lungs’. This is also repeated in the introduction and the conclusion.
Typo: should be ‘scarce’ not ‘scares’. Should be ‘broaden’ not ‘broader’
Page 2, Line 59 – should be ‘A difference’
Page 2 line 84 – should be ‘When comparing gene expression inflammation significantly differed between DBD and DCD lungs’ And line 85 should be ‘of’ not ‘fo’
Page 4
This sentence is missing some punctuation (closed brackets) and doesn’t quite make sense. “In 2007 Gomez-de Antonio et al. [20] reported their experience including 17 out-of-hospital (non heart beating donor-NHBD-lung transplantations performed since 2002 to 2005.”
Page 4 line 117, should be ‘of’ not ‘for air embolism’
Page 5 line 123 ‘A few years later’
Page 5 line 148 – suggest ‘were considered acceptable’
Page 7 line 202. Please do not use the term ‘harvest’ it is considered offensive to donor families. Also ‘cooling’ spelt incorrectly.
Table 1
I wonder if move Canada below Spain, as you discuss Canada after Spain in the text.
Is it possible to join Table 1 and Table 2 together.
That way the reader can more easily compare.
If it is not possible, or makes things worse as too big, don’t worry. Just a thought.
Just minor typo and grammar correction needed.
Author Response
Reviewer #1
We thank the reviewer for his/her kind comments.
We agree with the reviewer that this sentence (disparities still exist due to logistical and organization factors) may be not so clear. We decided to delete it in the revised version of the manuscript. – but provide no evidence for this claim.
In the present manuscript we aimed at summarizing the potentials of uDCDs only lung, mainly based on evidence. We prefer not to do a “call to action” since it would be quite difficult considering differences among centers.
Minor comments
Typos have been corrected. We apologize.
Table 1: following the reivewer’s suggestions “Canada” has been moved below “Spain”
We tried to make Table 1 and 2 just one Table, but, as the reviewer supposed, it was too big. So we leave the two tables separately.

Reviewer 2 Report
Useful review of current literature.
This manuscript comprises a comprehensive review of the literature concerning the use of uncontrolled DCD lung donors.
The review is fully international and usefully points out differences in the technical approaches that exist.
Useful call to use the data available to harmonise the technical approaches via an evidence based project.
The only addition I would ask the authors to consider would be some more detail on the combination of DCD donation and preservation using warm perfusion vs cold storage and expand role of EVLP
minor editing by native english speaker would improve flow
Author Response
Reviewer #2
We thank the reviewer for his/her comments.
Following the reviewer’s suggestion, we added a paragraph on hot topic of ongoing research on machine perfusion in lungs from uDCDs.
Typos have been corrected. We apologize.
